# Transplantation of Mature Adipocyte-Derived Dedifferentiated Fat Cells Facilitates Periodontal Tissue Regeneration of Class II Furcation Defects in Miniature Pigs

**DOI:** 10.3390/ma15041311

**Published:** 2022-02-10

**Authors:** Daisuke Akita, Tomohiko Kazama, Naoki Tsukimura, Yoshiki Taniguchi, Rie Takahashi, Yoshinori Arai, Niina Tsurumachi-Iwasaki, Hiroyasu Yasuda, Takahisa Okubo, Koichiro Kano, Taro Matsumoto, Masaki Honda

**Affiliations:** 1Department of Partial Denture Prosthodontics, Nihon University School of Dentistry, Tokyo 101-8310, Japan; akita.daisuke10@nihon-u.ac.jp (D.A.); tsukimura.naoki@nihon-u.ac.jp (N.T.); yasuda.hiroyasu@nihon-u.ac.jp (H.Y.); ookubo.takahisa@nihon-u.ac.jp (T.O.); 2Department of Functional Morphology, Division of Cell Regeneration and Transplantation, Nihon University School of Medicine, Tokyo 173-8601, Japan; kazama.tomohiko@nihon-u.ac.jp (T.K.); matsumoto.taro@nihon-u.ac.jp (T.M.); 3Section of Laboratory Animals, Nihon University School of Medicine, Tokyo 173-8601, Japan; taniguchi.yoshiki@nihon-u.ac.jp (Y.T.); takahashi.rie@nihon-u.ac.jp (R.T.); 4Department of Oral and Maxillofacial Radiology, Nihon University School of Dentistry, Tokyo 101-8310, Japan; arai.yoshinori@nihon-u.ac.jp; 5Department of Orthodontics, Nihon University School of Dentistry, Tokyo 101-8310, Japan; tsurumachi.niina113@gmail.com; 6Laboratory of Cell and Tissue Biology, College of Bioresource Science, Nihon University, Fujisawa 252-0880, Japan; kano.kouichirou@nihon-u.ac.jp; 7Department of Oral Anatomy, Aichi Gakuin University School of Dentistry, Nagoya 464-8650, Japan

**Keywords:** dedifferentiated fat cells, periodontal furcation defect, periodontal tissue regeneration, transplantation

## Abstract

Adipose tissue is composed mostly of adipocytes that are in contact with capillaries. By using a ceiling culture method based on buoyancy, lipid-free fibroblast-like cells, also known as dedifferentiated fat (DFAT) cells, can be separated from mature adipocytes with a large single lipid droplet. DFAT cells can re-establish their active proliferation ability and transdifferentiate into various cell types under appropriate culture conditions. Herein, we sought to compare the regenerative potential of collagen matrix alone (control) with autologous DFAT cell-loaded collagen matrix transplantation in adult miniature pigs (microminipigs; MMPs). We established and transplanted DFAT cells into inflammation-inducing periodontal class II furcation defects. At 12 weeks after cell transplantation, a marked attachment gain was observed based on the clinical parameters of probing depth (PD) and clinical attachment level (CAL). Additionally, micro computed tomography (CT) revealed hard tissue formation in furcation defects of the second premolar. The cemento-enamel junction and alveolar bone crest distance was significantly shorter following transplantation. Moreover, newly formed cellular cementum, well-oriented periodontal ligament-like fibers, and alveolar bone formation were observed via histological analysis. No teratomas were found in the internal organs of recipient MMPs. Taken together, these findings suggest that DFAT cells can safely enhance periodontal tissue regeneration.

## 1. Introduction

Periodontitis is a bacteria-induced chronic inflammatory condition, which leads to the destruction tooth-supporting structures and subsequent tooth loss [1,2]. Periodontitis not only leads to functional problems, compromising mastication, pronunciation, and sensation, but also has adverse effects on dental aesthetics. Recent studies reported that periodontitis incidence is increasing not only in the elderly, but also among younger individuals due to genetic, environmental, and diet-related factors [3]. Periodontal regeneration is a demanding procedure that requires restoration of the structure and function of the periodontal ligament, cementum, and alveolar bone. Appropriate cells, signaling factors, scaffolds, blood supply, mechanical loading, and microbial control are essential for creating optimal conditions for successful periodontal tissue reconstruction [4,5]. Albeit guided tissue regeneration (GTR) using membranes which prevent ingrowth of connective and epithelial cells during the initial wound healing phase is a well-established procedure, the treatment of intrabony defects remains a challenge and novel therapeutic strategies are needed [6]. Various studies have demonstrated that mesenchymal stem cells (MSCs) possessing sustained self-renewal and multi-lineage differentiation capability derived from the periodontal ligament, bone marrow, and alveolar periosteal can be effective for periodontal tissue regeneration in large animal models [7,8,9,10,11].

However, drawbacks related to the use of MSCs include low numbers of harvested cells and limited availability of harvested tissues. An ideal cell source for periodontal tissue reconstruction should have the characteristics of pluripotency, high proliferative ability, high purity, and ease of harvesting by dentists [12].

Adipose-derived stem cells (ASCs) has been demonstrated as a promising cell source for periodontal tissue engineering in rats, canine models, and other preclinical studies, highlighting their potential because adipose tissue contains considerable numbers of stromal cells that can be obtained through less invasive methods and in larger quantities when compared to bone marrow [13,14,15,16]. Mature adipocytes possessing a single, large lipid droplet are the most abundant cells in adipose tissue. We previously established lipid-free fibroblast-like cells and named these as dedifferentiated fat (DFAT) cells, obtaining them through the asymmetrical division of mature adipocytes via the ceiling culture method, which relies on the inherent buoyancy of adipocytes without the addition of any specific factors [17,18,19]. Unlike terminally differentiated adipocytes, DFAT cells exhibited a higher stable proliferation capability. DFAT cells have the characteristics of MSCs, with the multipotent capacity to differentiate into adipocytes [20,21,22], osteoblasts [20,21,23], chondrocytes [20], skeletal myocytes [24], smooth muscle cells [25,26], cardiomyocytes [27], and endothelial cells [28,29]. ASCs and DFAT cells can be isolated from subcutaneous adipose tissues or buccal fat pads [14,20,30,31,32,33]. Recent oral and maxillofacial research has demonstrated that DFAT cells and ASCs potentially contribute to the generation of skeletal bone and periodontal tissue in animals [14,30,32,34]. Based on our previously findings, the osteogenic potential of DFAT cells is higher than that of ASCs in vitro and in vivo [30,34,35]. Additionally, our previous work on periodontal regeneration demonstrated that rat DFAT cells have greater capacity for cementum and alveolar bone formation than ASCs [30]. However, it remains unclear whether DFAT cells can promote the reduction of junctional epithelium downgrowth and enhance alveolar bone formation in mandibular alveolar bone with three-wall defects. We later confirmed the efficacy of DFAT cells in reducing gingival recession and the formation of junctional epithelium in a three-wall intrabony periodontal defect model of rat alveolar bone. Taken together, our previous work has highlighted the potential of DFAT cells as a promising cell source for periodontal tissue regeneration. However, rats are not a reliable animal model as they have very high bone regeneration capacity. It is therefore imperative that our findings are validated in a large animal model prior to clinical research.

Adult miniature pigs (microminipigs; MMPs) are considered a useful model for biomedical research, especially in regenerative medicine, owing to their anatomical, physiological, and immunological similarities to humans [36,37]. In particular, MMPs are extensively used for xenotransplantation research because of their smaller body size and well-defined genetic background as compared to pigs [38,39]. Several studies have concluded that MMPs are an excellent translational model for autologous stem cell transplantation [40,41]. Further, MMPs are extensively used in regenerative dentistry research, including studies of periodontal regeneration [10,42,43,44,45,46,47,48]. Hence, we compared the regenerative potential of collagen matrix alone with porcine MMP-derived DFAT cell-loaded collagen matrix transplantation in the furcation defects and examined their safety in MMPs.

## 2. Materials and Methods

### 2.1. Experimental Animals

All animal experiments were reviewed and approved by the Animal Research and Care Committee of Nihon University (AP15M022 and AP19MED014). MMPs were purchased from Fuji Micra Inc. (Yamanashi, Japan) via Tokyo Laboratory Animals Science Co., Ltd. (Tokyo, Japan). According to previously published studies, six healthy adult MMPs (27.4 ± 1.5-month-old; 22.5 ± 4.0 kg) were purchased and examined for good systemic and oral health before the experiment [49].

### 2.2. Isolation and Culture of DFAT Cells

DFAT cells were established from fat tissue via a previously described method, with modifications [20]. In brief, approximately 1 g of abdominal subcutaneous fat tissue was washed extensively with phosphate-buffered saline (PBS; Wako, Osaka, Japan), minced, and digested in 0.1% (*w*/*v*) collagenase solution (C6885; Sigma-Aldrich, St. Louis, MO, USA) at 37 °C for 0.5 h with gentle agitation. The floating primary mature adipocytes in the top layer were collected after filtration and centrifugation at 700 rpm for 1 min. After three washes with PBS, 5 × 10^4^ cells were placed in 25 cm^2^ culture flasks (BD Falcon, England) filled completely with Dulbecco’s modified Eagle’s medium (DMEM; Sigma-Aldrich, Gillingham, UK) supplemented with 20% Fetal Bovine Serum (FBS; Sigma-Aldrich, Lot 14A189) and were incubated at 37 °C in 5% CO_2_ (Figure 1a). Mature adipocytes floated and adhered to the top inner ceiling surface of the flasks. During ceiling culture, fibroblast-like cells were generated from adhered adipocytes and formed colonies (Figure 1b). After about a week, the flasks were inverted, the medium was changed with DMEM supplemented with 20% FBS, and the flasks were inverted so that the cells were on the bottom. Media replacement was performed every 3–4 days. DFAT cells were used at the third passage. 1 × 10^6^ DFAT cells were transferred onto a square collagen matrix (Colla Tape; Zimmer Dental Inc., Carlsbad, CA, USA) resized to approximately 5 × 10 × 0.3 mm^3^, soaked in medium, incubated at 37 °C in 5% CO_2_, and prepared for transplantation.

### 2.3. In Vivo Experiment

The in vivo experiment was performed as described previously, with minor modifications [49,50], and is summarized in Figure 2. After emplacement (“−8 w”; Figure 3a), all MMPs were bred in the experimental animal facility at Nihon University School of Medicine (Tokyo, Japan) according to the guidelines of the Animal Research and Care Committee at Nihon University (AP15M022 and AP19MED014). All surgical procedures were performed under general anesthesia via intravenous injection of isoflurane (Pfizer Inc., NY, USA) and midazolam (Sandoz K.K., Tokyo, Japan), in addition to a local anesthetic of 2% lidocaine hydrochloride and 1:80,000 epinephrine (Nipro Co., Ltd., Osaka, Japan).

All calculus of the mandibular premolars was removed using a scaler (Hu-Friedy, Chicago, IL, USA) as initial periodontal therapy (“−6 w”; Figure 3b–d).

About 2 weeks later (“−4 w”), the first clinical assessments of probing depth (PD) and clinical attachment level (CAL) was performed in the partial regions (mesial, central, and distal) separated in the second premolar using a periodontal probe (Sun dental Co., Ltd., Osaka, Japan). The periosteum covering the buccal surface of the mandible was exfoliated after incisions were made along the superior border of the bilateral mandibular premolars. The alveolar bone overlying the furcation of the mandibular second premolar was removed using a water-cooled inverted dental bur. The furcation bony defect was approximately 4 mm wide, 5 mm deep, and extended 3 mm horizontally into the buccal furcation (Figure 3e,f). Notch-shaped marks were made on the root surface at the top of the alveolar crest and the floor of the defect. After creation of the alveolar bone defect, a silicone rubber impression material (EXAFINE; GC Dental Product Co., Ltd., Tokyo, Japan) was filled into the defect to induce chronic inflammation (Figure 3g). The flaps were repositioned and sutured using synthetic absorbable sutures (VICRYL; Ethicon Inc., NJ, USA). All MMPs were given the analgesic (loxoprofen sodium; Daiichisankyo Co., Ltd., Tokyo, Japan) after the surgery.

Four weeks after the initial surgery, the created furcation defects were randomly assigned to the control or DFAT group, and the clinical assessments of PD and CAL were recorded again before the second surgery (“0 w”; Figure 3h). Intrasulcular incisions were performed bucally to expose the root furcation defects. After periodontal debridement, the defects and roots of the second premolar were scaled with a scaler and washed with saline (Figure 3i). The right and left sides were randomly assigned to collagen matrix alone (*n* = 6) or DFAT cell-loaded collagen matrix (*n* = 6). The matrix with or without DFAT cells was tucked into the defects and covered with a membrane (BIOMEND, Zimmer Dental Inc.). The flaps were sutured using antibacterial absorbable sutures (VICRYL Plus; Ethicon Inc.) and fixed onto the wound region using periodontal dressing (COE-PAK, GC Dental Product Co., Ltd., Tokyo, Japan).

Twelve weeks after transplantation (“+12 w”), the third clinical assessments of PD and CAL were recorded before the bilateral mandibles of MMPs were extracted (Figure 3j). Organs, including the heart, liver, pancreas, and kidney, were then extracted and sliced to determine whether teratomas had formed. All six MMPs were subjected to the same clinical protocol and assessment.

### 2.4. Micro-Computed Tomography (CT) Imaging and Analysis

Micro-CT (R_mCT; Rigaku Corporation, Tokyo, Japan) was used as previously described [14,51]. The exposure parameters were 120 s, 90 kV, and 100 μA. 3D images were constructed using an image and data filing system (i-VIEW; J. Morita Co., Kyoto, Japan). The height from the root apex to the alveolar crest in the alveolar bone on the lingual side and the buccal side as well as the distance from the cemento-enamel junction (CEJ) to the alveolar crest were measured by radiologist in the partial regions (mesial, furcation, and distal) of the second premolar using bone volume measurement software 3by4viewer2011 (Kitasenjyu Radist Dental Clinic i-View Image Center, Tokyo, Japan).

### 2.5. Histological and Immunohistochemical Staining and Analysis

After CT analysis of the second premolar, the specimens were decalcified in 10% ethylenediaminetetraacetic acid (EDTA) for 5 weeks, dehydrated through a graded series of ethanol solutions, and then embedded in paraffin. For the specimens, sagittal plane sections (5 µm thick) were prepared with a microtome, and paraffin sections of the second premolar were stained with hematoxylin and eosin (H&E) and Azan to evaluate newly formed alveolar bone, cementum, and periodontal ligament in the furcation defect. For immunohistochemistry, paraffin-embedded specimens were deparaffinized and rehydrated with serial xylene and ethyl alcohol. After quenching endogenous peroxidase activity, the slides were rinsed three times in phosphate-buffered saline (PBS) for 5 min each. Cathepsin K primary antibody (Takarabio, Shiga, Japan) or periostin primary antibody (Abcam, Cambridge, MA, USA) was diluted 1:200 and incubated with samples at 4 °C overnight. Samples were then washed and incubated for an hour at room temperature. After rinsing three times in PBS (5 min each), samples were soaked in polymer reagent (EnVidion plus; Dako, Tokyo, Japan) for 30 min at room temperature for signal detection. All images were captured using a fluorescence microscope (BZ-X710; KEYENCE, Osaka, Japan).

For the quantitative analysis, three H&E-stained sections per specimen were selected, and the area of defect region, length of the epithelium legs from interradicular to notch-shaped marks, and height from the top of the newly formed alveolar crest to notch-shaped marks were imaged by pathologist under a fluorescence microscope (BZ-H3; KEYENCE, Osaka, Japan).

### 2.6. Statistical Analysis

Data are expressed as the mean and standard deviation (SD) for each group. Statistical analysis was performed using Excel (ystat2008.xls; Igakutosho-shuppan Ltd., Tokyo, Japan). The Mann-Whitney U test with Bonferroni correction was used for intergroup comparisons. Statistical significance was set at *p* < 0.05.

## 3. Results

### 3.1. Generation of Furcation Defects and Clinical Assessment in the Adult MMPs

After the initial surgery, inflammation-inducing periodontal class II furcation defects were successfully induced in the second premolars of the bilateral mandible at 0 weeks. Gingival swelling and bleeding on probing were observed in all MMPs before transplantation (Figure 3h).

The clinical parameters for each MMP are shown in Figure 4. In the control and DFAT groups, obvious reductions in PD were observed between 0 and +12 weeks in the central region (Figure 4a). Additionally, a significant difference was observed between the control and DFAT groups in the central region of the second premolar at +12 weeks (*p* = 0.0024). Clinical assessments of CAL indicated no significant difference between 0 and +12 weeks in the control group (Figure 4b). On the other hand, a significant reduction was observed between 0 and +12 weeks in the central region of DFAT group MMPs (*p* = 0.0279). Representative mean values and standard deviation (SD) of the PD and CAL at 0 and +12 weeks were shown in Figure 4c.

### 3.2. CT Analysis

To evaluate the regenerated bone volume and bone quantity, reconstituted 3D images, including frontal, sagittal, and horizontal sections of the second premolar at +12 weeks were generated by radiologist using i-VIEW (Figure 5).

Reconstituted 3D images of the control group revealed that the alveolar crest surface on the buccal side had an uneven surface and was downward relative to the cemento-enamel junction (CEJ) of the second premolar (Figure 5a). The frontal plane image of the distal root revealed that the distance from the root apex to the alveolar crest of the buccal side was shorter than that of the lingual side. The frontal plane image of the furcation also revealed that the distance from the alveolar crest to the tooth axis on the buccal side was lower than that on the lingual side. The alveolar crest surface on the buccal side was uneven. The frontal plane image of the mesial root revealed that the height from the root apex to the alveolar crest on the buccal side was lower than that on the lingual side. Artificially created notch-shaped marks were clearly visible in the sagittal plane image. The horizontal plane image of the mandibular premolars revealed no hard tissue formation on the buccal side of the mesial and distal roots.

In the DFAT group, reconstituted 3D images revealed that the alveolar crest surface on the buccal side was flat, and the alveolar crest was lower than the CEJ of the second premolar (Figure 5b). The frontal plane image of the furcation revealed that the alveolar bone on the buccal side was contiguous from the lingual side, and the alveolar crest had a smooth surface. The frontal plane image of the mesial and distal roots revealed that the height from the root apex to the alveolar crest of the buccal side was lower than that of the lingual side. In the sagittal plane image, the artificially created furcation defect was clearly visible, and notch-shaped marks were observed, similar to the control group. The horizontal plane image of the mandibular premolars revealed nebulous hard tissue formation around the buccal side of both roots in the second premolar.

To compare the dimensions from the root apex to the alveolar crest between the lingual and buccal side, bone volume-measuring software was used (Figure 6a). The ratio of the dimension of alveolar bone between the buccal and lingual sides in the control group was approximately 72–74%, and that in the DFAT group was approximately 84–86%. In the mesial and distal roots, the ratio of alveolar bone dimension in the DFAT group was significantly higher than in the control group. The distance from the CEJ to the alveolar crest was measured and is presented in Figure 6b. There were no significant differences in the mesial and distal regions between the control and DFAT groups. The distance from the CEJ to the alveolar crest in the furcation of second premolars in the control group was significantly longer than that in the DFAT group.

### 3.3. DFAT Cell Transplantation Enhanced Periodontal Tissue Regeneration

Photomicrographs of H&E-stained furcation defects in the control and DFAT groups are presented in Figure 7. The cavity in artificially created defects and notch-shaped marks made during the initial surgery were observed in both the control and DFAT groups. In addition, invasion of epithelium-like tissue into the alveolar crest was observed in both groups. In the control group, the distance of epithelium-like tissue invasion was approximately 4500 µm from the interradicular to notch-shaped marks (Figure 7a). The dented alveolar crest was observed approximately 1400 µm below the notch in the defect. In the DFAT group, invasion of the epithelium-like tissue continued approximately 2000 µm from the interradicular region (Figure 7b). The raised alveolar crest was approximately 1200 µm lower than the notch in the defect. The defect area in the control group was larger than that in the DFAT group (Figure 7c). A higher magnification of the interradicular region in Figure 7 is presented in Figure 8. In both groups, stratified squamous epithelia along with the root dentin without cementum-like tissues were observed in the yellow dotted line (Figure 8a,b). Epithelial invasion reached the notch-shaped marks in the control group but was suppressed only to the interradicular region (Figure 8c). The length of the epithelium in the control group was longer than that in the DFAT group (Figure 8d). The magnification of notch-shaped marks in Figure 7 is presented in Figure 9. In the control group, an eosinophilic cementum-like structure was observed approximately 300 µm below the notch of the defect area (Figure 9a). At higher magnification of the dotted frame, immunocytochemistry photomicrographs revealed no cementoblasts on the dentin of the created defect (Figure 9b). In the DFAT group, H&E-stained photomicrographs indicated that the eosinophilic cellular structure continued on the denuded dentin in the notch-shaped marks. At higher magnification of the dotted frame, immunohistochemistry revealed cylindrical periostin-positive cells located on the newly formed cellular structure (Figure 9d). The magnified root surface regions in Figure 7 are presented in Figure 10. H&E-stained photomicrographs of both the control and DFAT group revealed collagen bundles, blood vessels, and fibroblast-like cell structures between the root surface and alveolar bone (Figure 10a,e). A bone marrow-like structure was observed only in the alveolar crest. At higher magnification of the dotted frame, Azan staining photomicrographs revealed a bunch of collagen bundles in connective tissue between the root surface and alveolar bone surface in both the control and DFAT group (Figure 10b,f). Collagen bundles were observed between cementoblast-like cells near the newly formed cementum-like tissue on the dentin root surface. In control group, the bundles were inserted into cementum-like tissue (Figure 10b). However, invasion of collagen bundles was not observed in the root dentin. In DFAT group, these collagen bundles were observed between cells near the newly formed hard tissue on the dentin root surface and inserted into the ivory tube via the canaliculus in cementum (Figure 10f). H&E staining photomicrographs revealed that spherical cells were located in both the control and DFAT group on the eosinophilic hard tissue formation, including hematoxylin-positive cells (Figure 10c,g). At higher magnification of the dotted frame in the alveolar crest, osteocyte-like cells with nuclei and blood vessel-like structures in the lamellar-like hard tissue formation were observed in both the control and DFAT group (Figure 10d,h). A higher magnification of the alveolar crest in Figure 7 is presented in Figure 11. In both the control and DFAT group, HE staining and immunohistochemistry of the alveolar crest revealed cathepsin K-positive multinucleated osteoclasts on the irregular and ruffled lacunae.

The results of histometric analysis are shown in Figure 12. The height from the top of the newly formed alveolar crest to notch-shaped marks in the DFAT group was higher than that in the control group.

To investigate whether teratomas had formed, the organs were cut after extraction. Teratomas were not found in the internal organs, including the heart, liver, pancreas, and kidney (Figure 13a,d).

## 4. Discussion

In our previous study, we observed in vivo periodontal tissue regeneration, including cementum, periodontal ligament, and alveolar bone, after the allogeneic transplantation of DFAT cells from inbred rats into periodontal fenestration defects [30,32]. Moreover, transplanted fluorescently-labeled DFAT cells were observed in the periodontal ligament adjacent to newly formed bone and cementum [30]. However, there are no published studies on autologous DFAT cell transplantation for the regeneration of periodontal furcation defects in large animals or humans.

Adult MMPs are generally considered appropriate models for studying treatment modalities against human disease, including conditions in the oral maxillofacial region [36]. In the MMPs, swollen gingivae, accumulated plaque, and calculus bleeding on probing occur as gingivitis after the age of 6 months. Periodontitis is observed in MMPs after the age of 16 months, and the inflammatory process is similar to that observed in human periodontal disease as per histological observation [10,36]. In addition, serious periodontitis, including crestal resorption, can be caused via impression materials for alveolar bone destruction [49]. X-ray analysis indicated that the dentition of MMPs was diphyodont, changing from deciduous to permanent teeth after the age of 17 months [37]. Based on these studies, we employed healthy adult MMPs over the age of 17 months and used impression material to induce chronic inflammation.

In the control group, localized clinical attachment gain was observed at +12 weeks based on the PD and CAL. Reconstituted CT images of second premolars revealed the absence of hard tissue formation in the created furcation defect, and the morphology of the alveolar crest on the buccal side was irregular. In addition, bone volume measurement software revealed that the ratio of alveolar bone dimension between the buccal and lingual sides in the control group was approximately 72–74% of the root length. Histological observation revealed that the stratified epithelium legs continued from the interradicular region to the notch-shaped marks in the defect. Additionally, cementum and periodontal ligaments, including collagen bundles, were observed only between the root dentin below the notch-shaped marks and the dented alveolar crest. Immunohistochemistry revealed no hard tissue formation, including cementoblasts, on the denuded dentin in notch-shaped marks. Osteoclasts were observed on Howship’s lacunae in the alveolar bone. These results suggest that the transplanted membrane suppressed invasion of the epithelium and led to localized improvement of periodontal inflammation. In this study, exiguous limited regeneration was performed owing to the lack of MSCs within periodontal tissue in the control group.

In the DFAT group, obvious reductions in PD and CAL between 0 and +12 weeks were noted in the central regions of the second premolar. In particular, significant differences in PD were observed in the central region of the second premolar at +12 weeks. These results indicated marked attachment gain after DFAT cell transplantation. Reconstituted CT images of the second premolar revealed no vertical bone resorption, and a well-regulated alveolar crest was observed on the buccal side. Bone volume-measuring software revealed that the ratio of alveolar bone dimensions between the buccal and lingual sides in the DFAT group was approximately 84–86% of the root length and higher than that in the control group. The distance between the CEJ and the alveolar crest in the furcation of the second premolar was significantly shorter than that in the control group. Histological observation and analysis revealed that stratified epithelium legs were observed only in the interradicular region. The space of the remaining defect area in the DFAT group was smaller than that in the control group, and the length of the epithelium in the DFAT group was shorter than that in the control group. In addition, a newly formed cellular cementum structure, including cementoblasts, was observed in the notch-shaped marks. Furthermore, cementum and periodontal ligament, including the collagen bundle, were observed between the dentin root surface and the raised alveolar crest. Immunohistochemistry revealed osteoclasts on Howship’s lacunae, suggesting that bone remodeling was still in progress. Histological analysis suggested that the newly formed alveolar crest in the DFAT group was greater than that in the control group. These results are in accordance with an earlier report on periodontal regeneration [52,53]. We previously reported that DFAT cells have a high capacity for osteoblastic differentiation as well as cementum and alveolar bone formation in vivo [30]. Therefore, we sought to examine the periodontal tissue regeneration potential of DFAT cells in a large animal model prior to further clinical research, as rats are not a reliable animal model with regard to tooth regeneration. These findings imply that DFAT cell transplantation may prevent epithelial invasion and accelerate periodontal wound healing, resulting in enhanced periodontal tissue regeneration.

We confirmed the safety of DFAT cell transplantation based on the absence of teratomas in the heart, liver, pancreas, and kidney at +12 weeks. We did not administer any immunosuppressant drugs to MMPs and did not clean their teeth after transplantation. The created furcation defects were not completely regenerated after treatment. Plaque accumulation in the mandible of MMPs was a plausible consequence of the induced inflammation. In the future, we need to reconsider our experimental approach, including the methods for transplantation and animal breeding. Further, although teratoma formation was not observed in any of the MMPs, safety should not be overlooked in future pre-clinical and clinical research on DFAT cell transplantation. At Nihon University School of Medicine (Tokyo, Japan), cultured DFAT cells exhibiting neovascularization were obtained from a patient, expanded *in vitro*, and then applied in clinical trials of critical limb ischemia [54].

In summary, we observed that DFAT transplantation improved attachment gain, with the regenerated periodontium closely resembling original architecture, including collagen fibers inserted into the newly formed cementum and newly formed bone. No studies have explored the potential of using DFAT cells for the treatment of periodontal furcation defects. Our findings clearly indicate DFAT cell transplantation induced regeneration in the furcation area.

Taken together, DFAT cells are a promising cell source for oral and maxillofacial tissue engineering, including the regeneration of the periodontium. However, the impact of the local microenvironment on DFAT cells and the mechanisms controlling differentiation into cementoblasts, periodontal ligament fibroblasts, and osteoblasts remain to be further explored.

## 5. Conclusions

The aim of this study was to assess the periodontal regenerative potential and safety of autologous DFAT cell transplantation into inflammation-inducing periodontal class II furcation defects in MMPs.

In conclusion, attachment gain, newly formed cellular cementum, well-oriented periodontal ligament-like fibers, and alveolar bone formation were observed in MMPs receiving DFAT cell transplantation for the induced class II periodontal furcation defects. No teratoma was noted in the internal organs of recipient MMPs. Taken together, the current findings suggest that DFAT cells can safely enhance periodontal tissue regeneration, incentivizing further research toward their clinical application.

## Figures and Tables

**Figure 1 materials-15-01311-f001:**
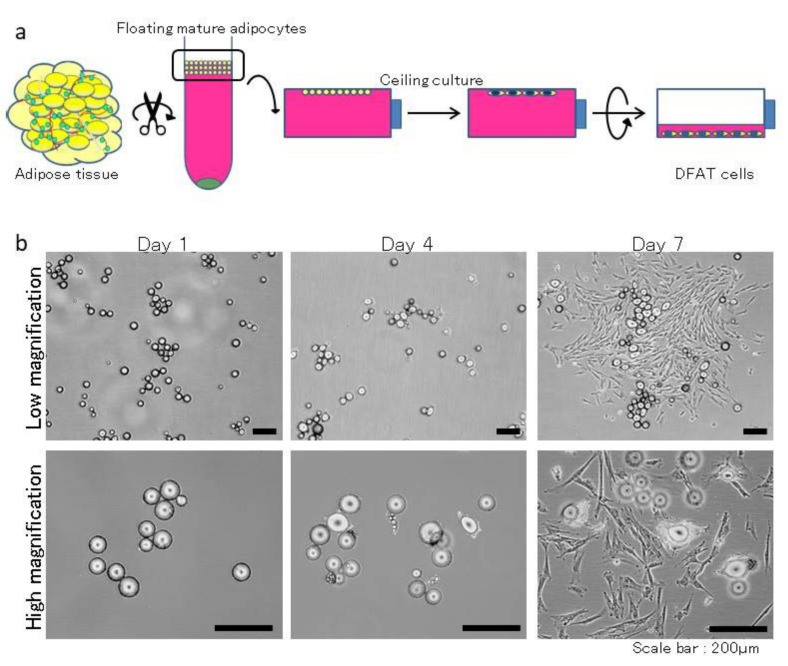
Isolation and morphology of DFAT cells. (**a**) After digestion and centrifugation of small pieces of fat tissue, the isolated unicolor mature adipocytes adhered to the top inner surface of a culture flask and generated fibroblast-like cells. (**b**) Microscopic view of primary cultured DFAT cells harvested from MMPs during the ceiling culture divide asymmetrically and generate fibroblast-like cells.

**Figure 2 materials-15-01311-f002:**
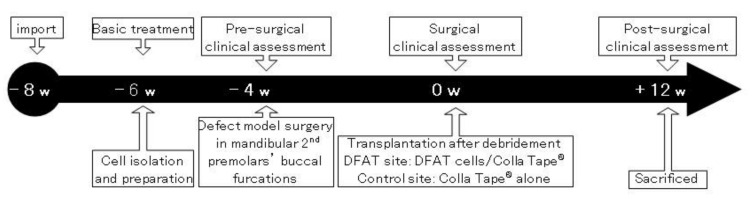
The timetable of in vivo experimental procedures. Six MMPs with 12 teeth in total were included.

**Figure 3 materials-15-01311-f003:**
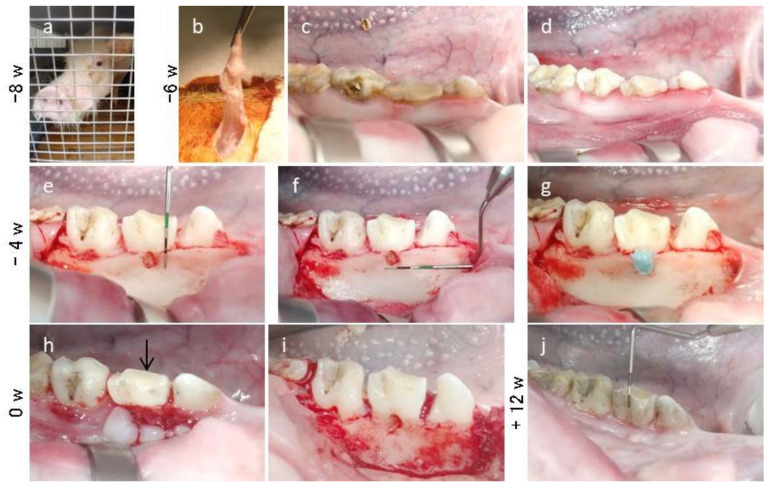
Creation of the periodontal furcation defect in the mandible of MMPs. (**a**) MMP at the Nihon University School of Medicine. (**b**) Small pieces of adipose tissue obtained from the hypogastrium. (**c**,**d**) Removal of the calculus in the supragingival region of mandible premolars using a curette. (**e**,**f**) The furcation defects (4 mm wide, 5 mm deep, and 3 mm horizontally) were created on the buccal side of the bilateral second premolars. (**g**) Filling the impression material into the created furcation defect to induce chronic inflammation. (**h**) Four weeks later, inflammation was observed at the buccal surface. Black arrow indicates gingival inflammation and bleeding from the buccal side of the bilateral second premolar. (**i**) After debridement, the furcation defect and the buccal bone of premolars exhibited destruction. (**j**) Twelve weeks later, the clinical parameters were once again measured before extracting the mandible.

**Figure 4 materials-15-01311-f004:**
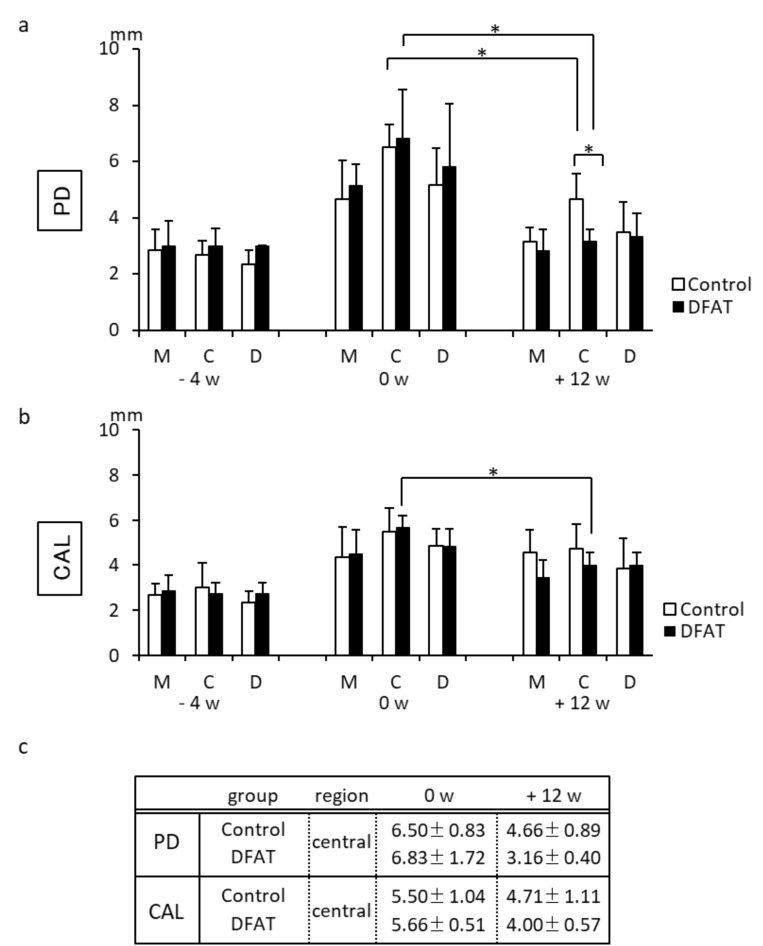
Clinical parameters at the buccal surface of the mandibular second premolar at three different time points. (**a**) In the control group, a significant difference was observed in the central area of the second premolar between 0 and +12 weeks. In the DFAT group, significant differences were observed in the whole area of the second premolar between 0 and +12 weeks. In addition, a significant difference was observed in the central area between the control group and DFAT group at +12 weeks. (**b**) A significant difference was observed in the central area of the DFAT group between 0 and +12 weeks. Each bar represents the mean ± SD (*n* = 6, * *p* < 0.05). M, mesial; C, central; D, distal. (**c**) Results of mean values (±SD) in central region of DFAT group and control group between 0 and +12 weeks (mm).

**Figure 5 materials-15-01311-f005:**
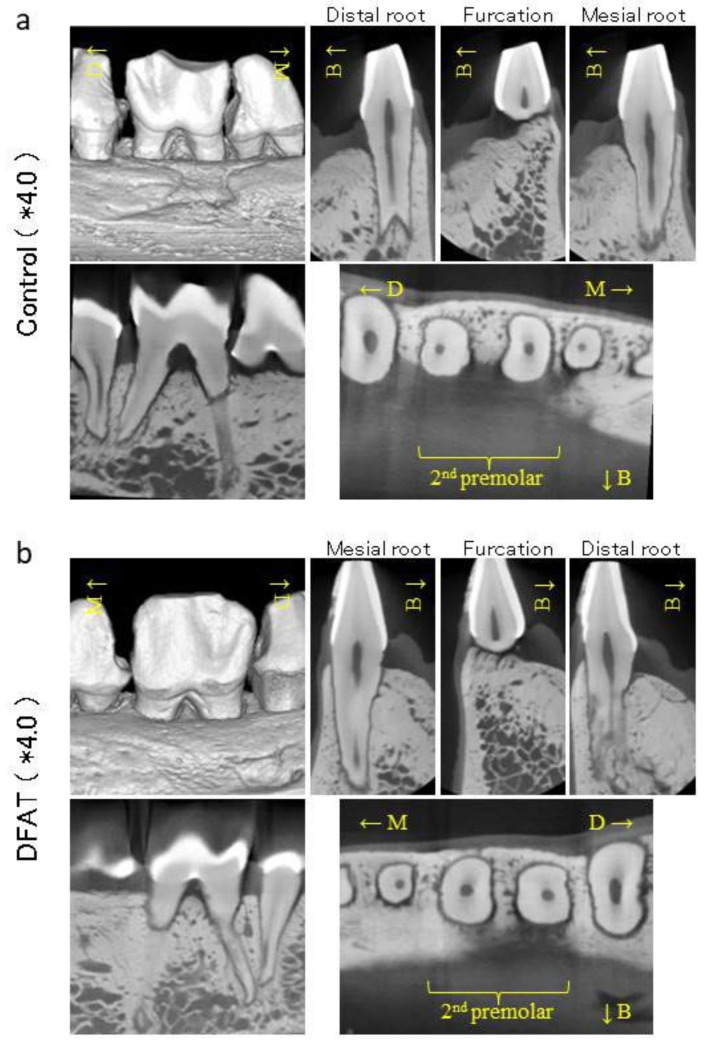
Representative micro-CT images of the mandibular second premolar at +12 weeks. (**a**) The upper left panels show CT images of the control group reconstructed using i-VIEW. Alveolar bone with irregularities was observed on the buccal side of mandibular. The upper right panel shows the frontal plane section of the second premolar. Frontal plane images revealed that the height of the alveolar bone from the root apex to alveolar crest on the buccal side was lower than that on the lingual side. The frontal plane image of the furcation revealed that the buccal side of the alveolar bone formation was uneven. The lower left panel shows the sagittal plane section. The artificially created furcation defect and notch are clearly visible. The lower right panel shows the horizontal plane section of the mandible. Hard tissue formation was not observed on the buccal side of mesial and distal roots of the second premolar. (**b**) The lower left panel shows CT images of the DFAT group reconstructed using i-VIEW. Alveolar bone without the step was observed on the buccal side of mandibula. The lower right panel shows the frontal plane section of the second premolar. The lower left panel shows the sagittal plane section. The artificially created furcation defect and notch are clearly visible. The lower right panel shows the horizontal plane section of the mandible. Inarticulate hard tissue was observed on the buccal side of mesial and distal roots of the second premolar. Asterisks indicate magnification.

**Figure 6 materials-15-01311-f006:**
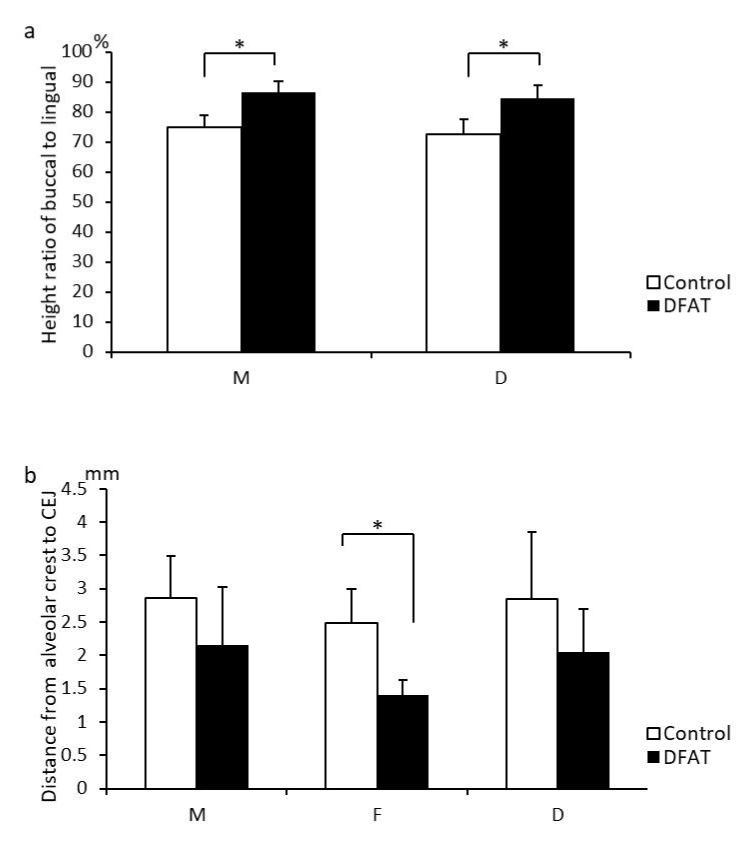
Quantification of the micro-CT analysis. (**a**) Comparison of the dimensions from the root apex to the alveolar crest in the alveolar bone between the lingual and buccal sides. The ratio of the dimension at mesial and distal roots in the DFAT group was greater than that in control group. (**b**) Dimension from the alveolar crest to the cemento-enamel junction. A significant difference was observed in the furcation between the control group and DFAT group. Each bar represents the mean ± SD (*n* = 6, * *p* < 0.05). M, mesial; F, furcation; D, distal.

**Figure 7 materials-15-01311-f007:**
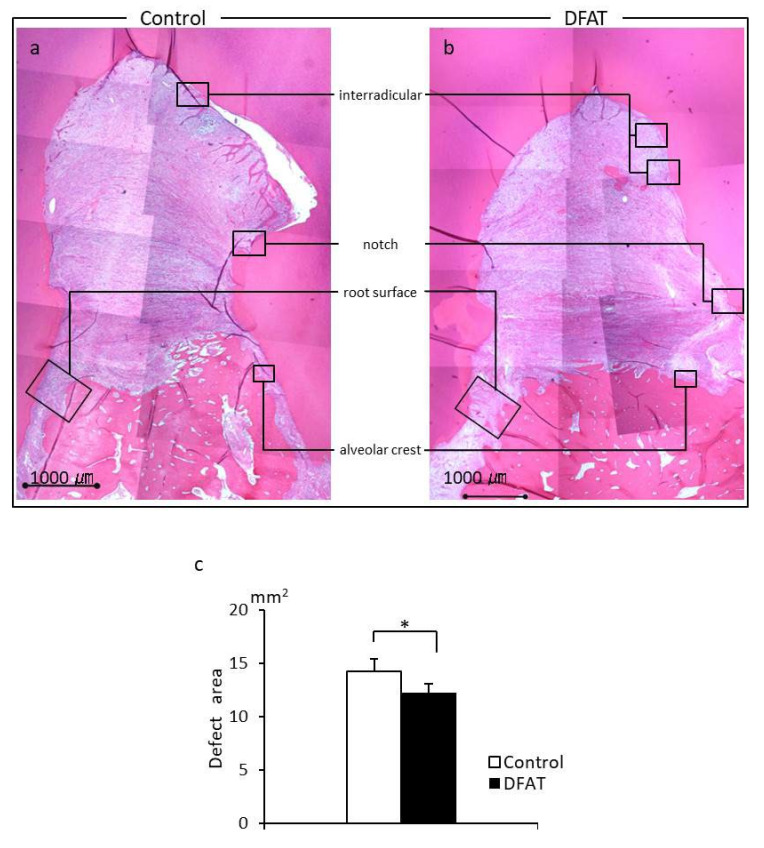
Representative H&E-stained sagittal plane section from the furcation of the second premolar at +12 weeks. Furcation defects in the control group (**a**) and DFAT group (**b**) were easily identified in the central part of the image. (**c**) Quantification of the defect area. The defect area was smaller in the DFAT group that in the control group. Each bar represents the mean ± SD (*n* = 6, * *p* < 0.05).

**Figure 8 materials-15-01311-f008:**
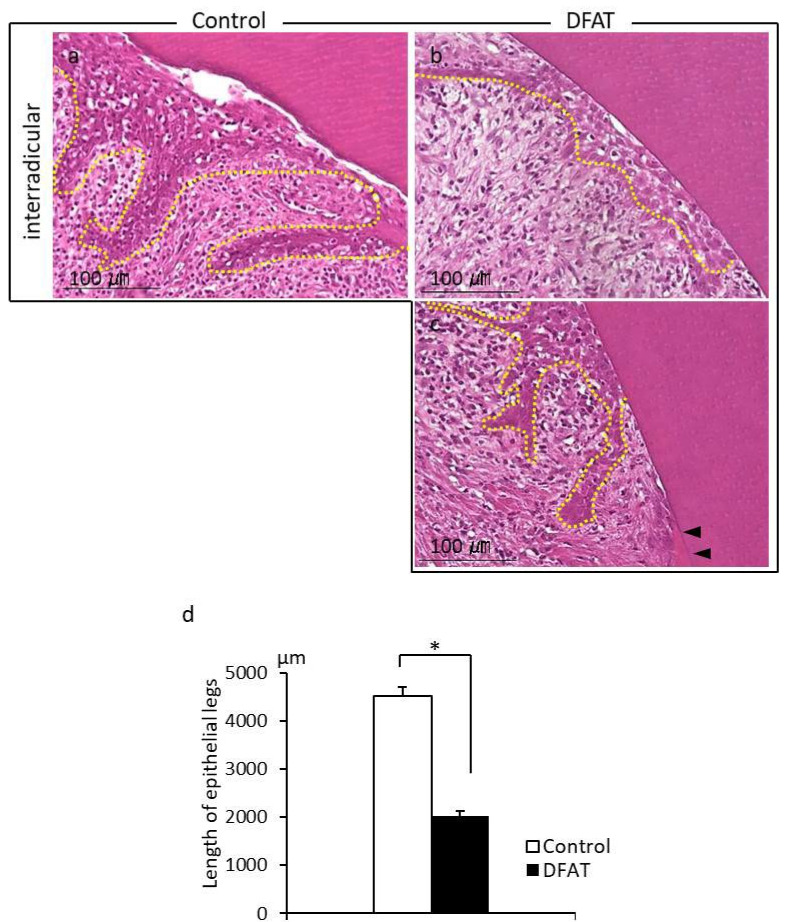
High-magnification image of interradicular regions from Figure 7. (**a**) Control group; the dotted line indicates epithelial legs reaching the upper section of the furcation defect. (**b**) DFAT group; the dotted line indicates epithelial tissue in the upper section of the furcation defect. (**c**) The dotted line indicates epithelial legs above the hard tissue formation. Black arrowheads indicate newly formed cementum. (**d**) The length of epithelial legs was longer in the control group than in the DFAT group. Each bar represents the mean ± SD (*n* = 6, * *p* < 0.05).

**Figure 9 materials-15-01311-f009:**
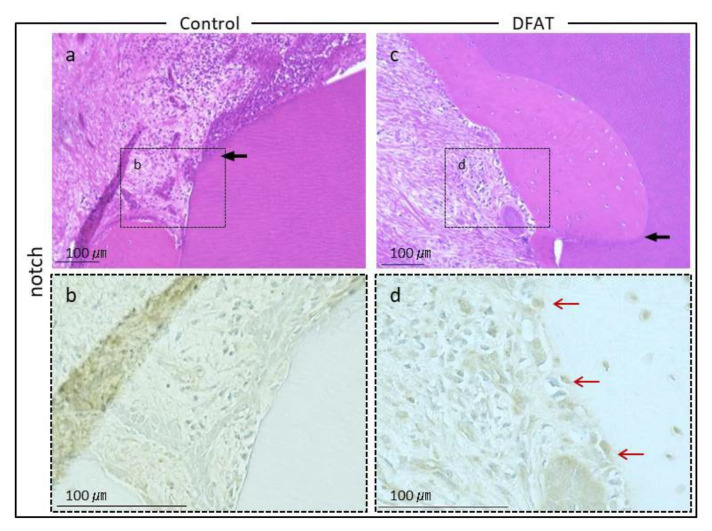
Magnified notch regions of Figure 7. (**a**) In the control group, epithelial tissue was observed above the notch of the created defect (black arrow). (**b**) At higher magnification of the dotted frame in (**a**), immunocytochemistry revealed no cementoblasts in the created defect. (**c**) In DFAT group, defect sites and eosinophilic structures were observed within the created furcation defect (black arrow). (**d**) At higher magnification of dotted frame in (**c**), immunocytochemistry revealed periostin-positive cells in the newly formed cementum. Red arrows indicate the newly formed cementoblasts.

**Figure 10 materials-15-01311-f010:**
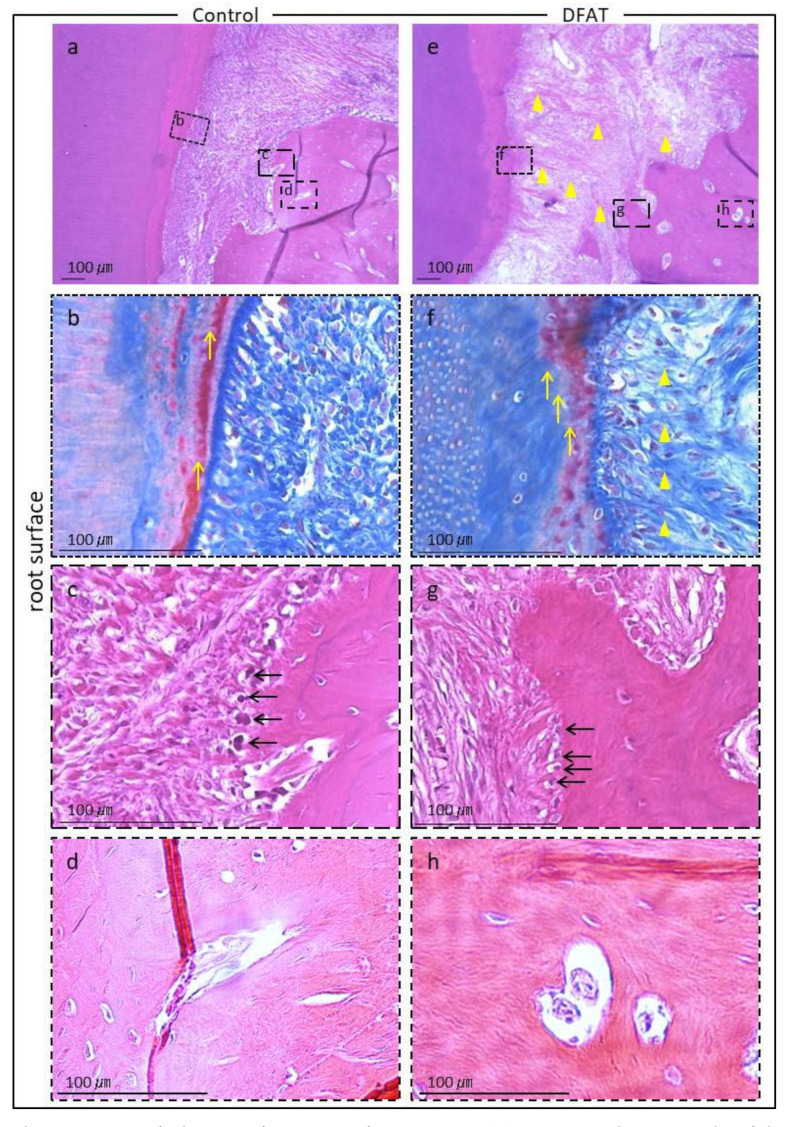
Magnified root surface regions from Figure 7. (**a**) H&E-stained micrographs of the cementum-ligament-alveolar bone complex from the control group. (**b**) At higher magnification of dotted frame in (**a**), azan-stained histological section of the dentin root surface exhibits cementum and periodontal ligament (in the central part of the image). Yellow arrows indicate Sharpey’s fiber. (**c**) At higher magnification of the dotted frame of alveolar bone surface in (**a**), osteoblasts lining the alveolar crest were identified. (**d**) At higher magnification of dotted frame in the alveolar crest in (**a**), the bone marrow-like organization, including blood vessel structures, was observed. (**e**) H&E-stained micrographs of the cementum-ligament-alveolar bone complex in the DFAT group. Yellow arrowheads indicate collagen bundles. (**f**) At higher magnification of the dotted frame in (**e**), azan-stained histologic sections harbor collagen bundles inserted perpendicular to the dental root surface. Yellow arrows indicate Sharpey’s fiber. Yellow arrowheads indicate collagen bundles. (**g**) At higher magnification of the dotted frame on alveolar bone surface in (**e**), osteoblasts lining the alveolar crest can be observed. (**h**) At higher magnification of the dotted frame in the alveolar crest in (**e**), the bone marrow-like organization, including blood vessel structure, can be observed.

**Figure 11 materials-15-01311-f011:**
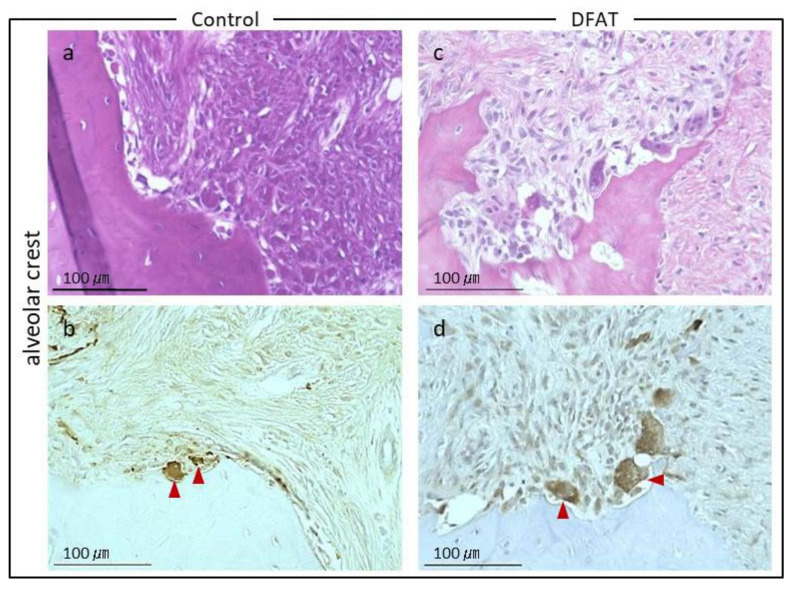
High-magnification image of alveolar crest regions from Figure 7. (**a**) H&E-stained micrographs from the control group. (**b**) Immunocytochemistry photomicrographs. Red arrows indicate cathepsin K-positive osteoclasts in the control group. (**c**) H&E-stained micrographs in the DFAT group. (**d**) Immunocytochemistry photomicrographs. Red arrowheads indicate cathepsin K-positive osteoclasts lining the newly formed alveolar crest in the DFAT group.

**Figure 12 materials-15-01311-f012:**
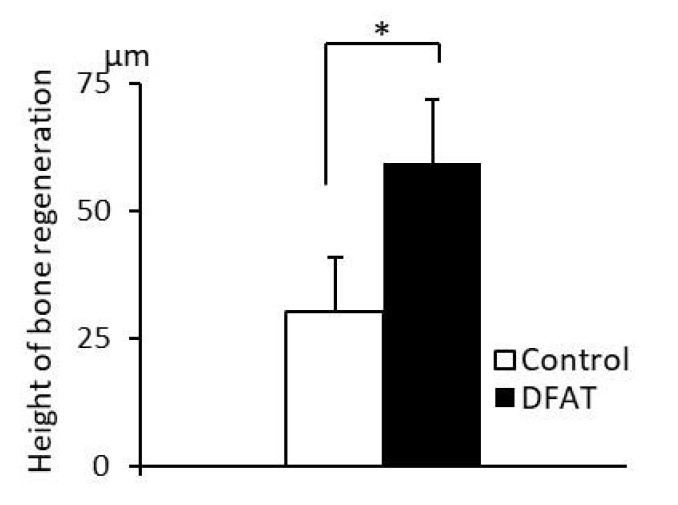
Histometric analysis of newly formed alveolar bone in the furcation of the second premolar at +12 weeks. The height from the top of the newly formed alveolar crest to notch-shaped marks in the DFAT group was higher than that in the control group. Each bar indicates the mean ± SD (*n* = 6, * *p* < 0.05).

**Figure 13 materials-15-01311-f013:**
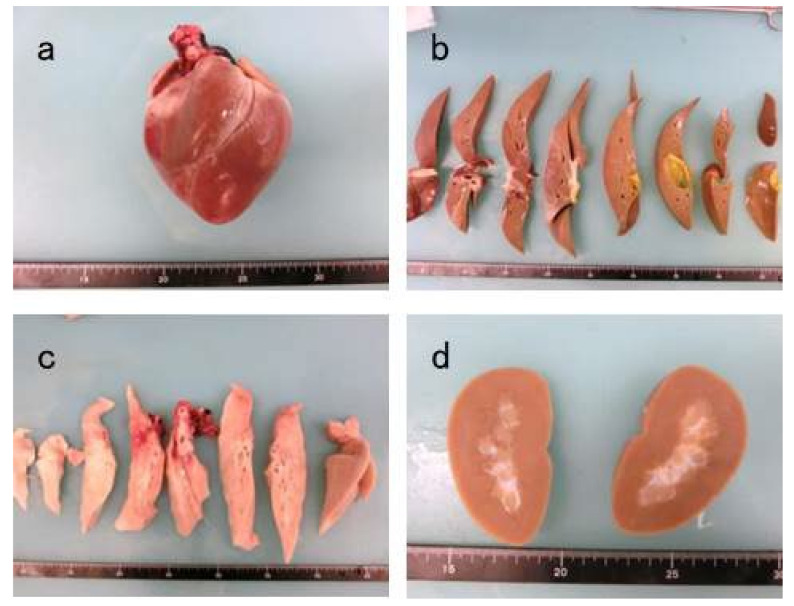
Representative observation of internal organs at + 12 weeks. (**a**) heart. (**b**) liver. (**c**) pancreas. (**d**) kidney.

## Data Availability

The data presented in this study are available in insert article.

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
