# Peer review of "Transplantation of Mature Adipocyte-Derived Dedifferentiated Fat Cells Facilitates Periodontal Tissue Regeneration of Class II Furcation Defects in Miniature Pigs"

_materials, 2022, doi:10.3390/ma15041311_

Round 1

Reviewer 1 Report

The aim of the presented manuscript „Transplantation of mature adipocyte-derived dedifferentiated fat cells facilitates periodontal tissue regeneration of class II  furcation defects in miniature pigs” was to evaluate the regenerative potential of autologous dedifferentiated fat (DFAT) cells cell transplantation in adult miniature pig.

Generally speaking, the manuscript is nicely constructed and very informative. The H&E stained pictures add significantly to the merit of the paper. I have some suggestions that I hope would be considered by the authors.

Abstract: why do not the authors mention that two sites (tests and controls) are being evaluated? 

Introduction:

1. Please add information about the prevalence of periodontitis according to recent reports.

2. Please explain the cornerstone in periodontal tissue regeneration, which is guided tissue regeneration with membranes. Recently, perforated membranes were used in clicnial studies to allow for better cellular and mediator interactions, as well as MSC from periosteal and gingival connective tissue involvement in regeneratory processess, both in chronic (DOI: 10.1902/jop.2012.120301) as well as in aggressive periodontitis (DOI: 10.1007/s00784-018-2368-3). In this matter, the application of MSCs represents another step in periodontal regeneration.

3. Please add information that MSCs can. Be derived also from gingiva (healthy as well as inflammed) and use appropriate citation, for example doi: 10.5604/17322693.1214383. There are some obvious advantages related with this partciulars cells.

The aim should mention that the comparison was made between the application of collagen matrix alone (controls) or DFAT cell-loaded collagen matrix (tests). 

Materials and methods are nicely described. My main objective is the relatively small number of evaluated sites (12 in total). Another issue is the lack of descirbed means. Authors used  bar graphs to show the mean values of evaluated parameters, but they are not clearly defined. I strongly believe that construction of table, in which means and SD of tests and controlas at baseline and at 12 weeks are depicted, will improve the general visibility of presented data.

In the discussion section the potential problems to solve, before MSCs can be introduced in the clinical practice, should be shortly mentioned.

Conclusions should correspond with the changed „aim of the study”.

Reviewer 2 Report

Authors investigated whether transplanting porcine MMP-derived DFAT cells would promote the regeneration of furcation defects and examined their safety in MMPs. They have done a relevant study meticulously and the results have been presented  well. 

Few minor corrections

Significant bars in Fig 4 is different as compared to Fig 6,7 & 8 . Uniformity should be maintained.

Reviewer 3 Report

The article entitled " Transplantation of mature adipocyte-derived dedifferentiated 2 fat cells facilitates periodontal tissue regeneration of class II 3 furcation defects in miniature pigs. " it is very interesting, well-founded, and well-justified as to its importance and necessity. However, the article is excessively long (especially in its results) and tiring. Despiste being excessively long it still leaves methodological gaps.

despite the introduction being very enlightening on the subject and justifying the research, it is excessively long. I suggest shortening/summarizing mainly in the part about the differences between ASC and DFAT. The third, fourth and fifth paragraphs can be summarized in just 1 paragraph that talks about the evolution, differences and superiorities between ASC and DFAT.

on what basis was the sample size defined? (6 in each group)

The present reviewer believes that a table with the results is necessary and that there is no statistical difference between the groups evaluated at baseline. Could be a supplemental table

The authors report on two occasions that the methodology was carried out according to previously published studies, WITH MODIFICATIONS. What modifications? these modifications must be very well described and clear to readers.

how was the collagen matrix loaded with DFAT cells?

clinical parameters were not evaluated immediately before surgery, as a negative control. This would help to show that there is no initial statistical difference between the evaluated groups. Although they were randomly distributed, equality between these clinical parameters at baseline is important to demonstrate.

Also, was any medication administered to the animals after intervention?

Was the person responsible for the clinical analysis the same person responsible for the surgery? Or was the evaluator blind? If not, How did the authors avoid introducing measurement and analysis bias?

Describe how the groups were randomized (method)? How was allocation concealment performed?

The authors report that they used only the mann-whitney test. This test is a test for independent nonparametric data. The application of this one to compare data between control group versus DFAT is adequate.

However, analyzes were performed comparing the same group at different times. For this comparison, a test for dependent samples must be applied, since they come from the same tooth/defect evaluated in more than one moment.

the differences between moments and between times are not clearly described in the graph. I would like the authors to provide a table with the values of each comparison.

The authors constructed excessively long paragraphs for descriptive results. Authors should summarize and include in the images the main findings Se the paragraph that goes from line 337 to 394!!!!!

it is observed in the CT scans that the root was still forming. Could this have influenced the pigs' response? it is known that, in teeth with incomplete root formation, the resorption process was faster. And that seems to be visible in the Micro-CT scans.

The discussion presents a lot of repetition of results, and little discussion with previous studies (except for studies by the team of authors).

Author Response

The reviewer pointed out "WITH MODIFICATIONS" means initial periodontal therapy and change of the impression material.

Reviewer 4 Report

Abstract :

Line 33 -  computed tomography (CT) - micro computed tomography

Material & Methods :

Line 208 - alveolar crest were measured in the partial - who performed the assessments ? - give the details

Why only a single evaluator was used for assessment ?

Was the assessor calibrated or qualified ?

Histology :

Line 218 - to evaluate  newly formed alveolar bone - again was the evaluator calibrated / blinded to groups & were multiple evaluators used - this could be a source of bias introduced

CT analysis :

Line 270 - To evaluate the regenerated bone volume -  again who did the evaluation & was he calibrated / blinded

Results :

Line 333 - DFAT cell transplantation enhanced periodontal tissue regeneration  - suggest ,this section can be rewritten in a more concise manner, it is a bit too long, with many similar sentences - to make it easier for the readers

Line 394 : why is there more  cathepsin  K-positive multinucleated giant cells (figure 11) in the DFAT group & what does this signify ?

Discussion :

Limitations of the study & the use of DFAT has to be mentioned 

Conclusions :

Line 545 - treatment of periodontal furcation defects - change to induced class ii periodontal furcation defects 
